# Evaluation of Transplacental Antibody Transfer in SARS-CoV-2-Immunized Pregnant Women

**DOI:** 10.3390/vaccines10010101

**Published:** 2022-01-10

**Authors:** Ching-Ju Shen, Yi-Chen Fu, Yen-Pin Lin, Ching-Fen Shen, Der-Ji Sun, Huan-Yun Chen, Chao-Min Cheng

**Affiliations:** 1Department of Obstetrics and Gynecology, Kaohsiung Medical University Hospital, Kaohsiung Medical University, Kaohsiung 807, Taiwan; 2Institute of Biomedical Engineering, National Tsing Hua University, Hsinchu 300, Taiwan; sandy216621@gmail.com (Y.-C.F.); peggy1240309@gmail.com (Y.-P.L.); 3Department of Pediatrics, National Cheng Kung University Hospital, College of Medicine, National Cheng Kung University, Tainan 704, Taiwan; drshen1112@gmail.com; 4Department of Obstetrics and Gynecology, Pojen Hospital, Kaohsiung 804, Taiwan; Gummysun@gmail.com; 5Department of Obstetrics and Gynecology, Kaohsiung Chang Gung Memorial Hospital, Kaohsiung 833, Taiwan; B101102082@tmu.edu.tw

**Keywords:** neutralizing antibody, COVID-19 vaccine, pregnancy, cord blood, Delta variant, maternal immunity

## Abstract

Background: Severe acute respiratory syndrome coronavirus 2 (SARS-CoV-2) infection during pregnancy could result in adverse perinatal outcome. Clinical data on the assessment of the immune response in vaccinated pregnant women and subsequent transplacental antibody transfer are quite limited. Objective: To assess maternal and neonatal neutralizing antibody levels against both wildtype and Delta (B.1.617.2) variants after maternal mRNA vaccination. Study Design: This cohort study was conducted 29 pregnant women who were vaccinated at least one dose of Moderna (mRNA-1273) vaccine. Both neutralizing antibody (wildtype and Delta variant) and S1 receptor binding domain IgG antibody levels were evaluated in maternal and cord blood on the day of delivery. Results: Superiority of antibody level was significant in fully vaccinated women compared with the one-dose group (maternal sera, median, 97.46%; cord sera, median, 97.37% versus maternal sera, median, 4.01%; cord sera, median, 1.44%). No difference in antibody level was noted in relation to interval of second immunization to delivery in the two-dose group (95.99% in 0–2 weeks, 97.45% in 2–4 weeks, 97.48% in 4–8 weeks, 97.72% in 8–10 weeks). The most pronounced reduction was observed for the Delta variant. The wildtype neutralizing antibody level of full-vaccinated women was not influenced by the pertussis vaccination. Conclusion: The data underscore the importance of full vaccination in pregnancy and support the recommendation of COVID-19 immunization for pregnant women. The lower level of vaccine-induced neutralizing antibodies for the Delta variant indicates insufficient protection for mother and newborn and highlights the need for development of effective vaccine strategies.

## 1. Introduction

In recent decades, the value of vaccinating pregnant women as a means of protecting both mother and fetus has been well documented [1]. After the mother is vaccinated, antibodies can be delivered before birth via the placenta and after birth via breastfeeding [1,2]. The fetal immune system is very different from the newborn immune system, and newborns are more susceptible to microbes and environmental damage [3,4]. Vaccination during pregnancy as a means of increasing maternal antibody levels and enhancing passive immunity in infants has been effective against some neonatal infections, such as tetanus [5,6]. How much antibody the mother can deliver to the newborn depends on the concentration of the mother’s antibodies, which is directly related to the time point at which the mother receives the vaccine [7].

Previous studies have shown that antibodies produced by mothers infected with COVID-19 pass through the placenta to the fetus [8,9]. Although the risk and symptoms of pregnant women infected with COVID-19 are the same as those of non-pregnant people [10,11], several studies have found that, compared with age-matched non-pregnant women, pregnant women are more likely to have severe symptoms after being infected with COVID-19, especially when combined with risk factors such as advanced age, preeclampsia, obesity, diabetes, and high blood pressure [11,12,13]. The risk of neonatal complications such as premature delivery, meconium staining, respiratory distress, and perinatal death increases as well [12,13,14,15,16,17,18,19,20]. According to the characteristics of different COVID-19 vaccines, the components in the mRNA vaccine would not enter the nucleus of the host cell and remain separated from the host DNA. The mRNA in the vaccine is decomposed by the host cell within a few days, so it is considered more suitable for vaccination during pregnancy [21]. Although a randomized controlled trial study on the safety and adverse reaction characteristics of COVID-19 vaccines did not include pregnant women, as of 4 October 2021, more than 163,000 pregnant women from different ethnic backgrounds in the United States have been vaccinated with either Pfizer-BioNTech (BNT162b2) or Moderna (mRNA-1273) vaccines with no evidence of harm [22,23], or an increase in pregnancy-related complications [24]. Studies have also shown that antibodies produced by mothers infected with COVID-19 during pregnancy can be found in newborn blood, umbilical cord blood, and breast milk, confirming the phenomenon of passive immunity [8,9]. Additionally, a Dutch study found that IgA antibodies in women’s breast milk were present for up to ten months after a confirmed maternal infection [25].

Although newborns rely on the transplacental transfer of IgG produced by their mothers to resist the COVID-19 virus, there is not sufficient evidence to support the report [26,27]. However, based on previous evidence regarding the vaccination of pregnant women, it should have a positive impact. The aim of this study is to assess the post-vaccination immune response (i.e., antibody production) in pregnant women and the correlation between immune response and vaccination time point. Due to increased concern regarding COVID-19 variants, we have included Delta (B.1.617.2) variant-associated results.

## 2. Materials and Methods

### 2.1. Study Design and Patients

This is a prospective study approved by institutional review board of Kaohsiung Medical University Hospital (IRB number: KMUHIRB-SV(II)-20210087). All participants were confirmed negative for SARS-CoV-2 infection with nasopharyngeal swab reverse transcription-polymerase chain reaction on the day of admission and received at least one dose of Moderna (mRNA-1273) SARS-CoV-2 vaccine. Results were obtained from the peripheral blood and umbilical cord blood of 29 mothers on the day of delivery. Newborn blood samples were collected from the umbilical cord after clamping. Of the participants, 25 completed two doses and four had one dose.

All patients in the study had singleton pregnancies without symptoms related to COVID-19 during pregnancy. They had voluntarily vaccinated against COVID-19 with first dose administration between the 27th and 38th week of gestation. The exclusion criteria were age below 20 years, COVID-19 vaccination before pregnancy, preterm labor, and disease with immunosuppressant treatment.

Regarding the protocols of measuring neutralizing antibody and S1 receptor binding domain IgG antibody, we have described the details in Appendix A.

### 2.2. Collection of Variables

The following data were obtained from the medical records: the mother’s age, weight, height, race, estimated date of confinement, parity, medical and obstetric history, date of COVID-19 vaccination, date of pertussis immunization, blood drawing dates, delivery mode, newborn’s birth date, sex, birth weight, and Apgar scores at one and five minutes. Maternal and cord blood neutralizing antibody levels were determined as dependent variables.

### 2.3. Statistical Analysis

The data were analyzed using GraphPad Prism. The correlation between the two different data, maternal and cord blood, were obtained using the Spearman rank correlation coefficient and a Bland–Altman plot. Results of *p* < 0.05 were considered to be statistically significant.

## 3. Results

Blood samples were collected from 29 mothers on the day of delivery and from the umbilical cord blood of 29 newborns. Of the pregnant vaccine recipients, 25 completed two doses and four had one dose. Of the four pregnant women who did not complete the two doses of the vaccine, three were due to the shortage of vaccines, and one was because the interval required for the administration was not reached.

### 3.1. Participant Characteristics

Participant demographic and clinical characteristics are presented in Table 1. All mothers confirmed negative for COVID-19 infection by nasopharyngeal swab reverse transcription-polymerase chain reaction results performed on the day of admission. The study population consisted of Asian women who were all vaccinated with the Moderna mRNA-1273 vaccine without severe side effects observed after administration. No maternal, obstetric, or neonatal complications were noted. Median maternal age was 33.21 years (IRQ 31–35) with a median gestational age of 38.55 weeks at the time of delivery. The median gestational age at the first vaccine dose was 28.45 weeks with one woman receiving their first dose in the second trimester, and 28 receiving their first dose in the third trimester.

### 3.2. Sample Characteristics

Median percentage of inhibition of neutralizing antibodies against wildtype SARS-Co-2 in maternal sera at the time of delivery was significantly lower in those vaccinated with one dose compared to those vaccinated with two doses (40.32% versus 97.46%, Table 2). For those women who received two-dose vaccinations, median percentage of inhibition of neutralizing antibodies in maternal sera did not differ in relation to the interval of second immunization to delivery (95.99% in 0–2 weeks, 97.45% in 2–4 weeks, 97.48% in 4–8 weeks, 97.72% in 8–10 weeks, respectively, Table 2). The result was similar in neonatal sera (Rho = 0.5681, *p* = 0.0031) and cord-to-maternal ratio (Rho = 0.2389, *p* = 0.2501). For women who received a one-dose vaccination, the median percentage of inhibition of neutralizing antibodies for SARS-CoV-2 Delta variant in maternal sera and cord blood were 4.01% and 1.44% (Table 2), which were below the cutoff value (30%) for presence of neutralizing antibodies [27]. In the two-dose immunization group, the inhibition percentage of maternal and cord blood neutralizing antibodies for the Delta variant group were 49.96% versus 41.86%, 80.72% versus 57.48%, 81.76% versus 70.60% at 0–2, 2–4, 4–8 weeks from second vaccination time point to delivery respectively (Table 2). The median cord-to-maternal ratio of Delta variant neutralizing antibodies in the two-dose group did not differ in regard to vaccination timing. The inhibition percentage of neutralizing antibodies for the Delta variant was significantly lower in maternal sera and cord blood compared with wildtype (*p* = 0.0001 & *p* < 0.0001, Figure 1). Among cases, the inhibition percentage of neutralizing antibodies for the Delta variant was 29.4% lower in maternal sera and 37.6% lower in cord blood. In the two-dose immunization group, all mother–neonatal pairs were positive for SARS-CoV-2 S1 receptor binding domain IgG antibodies, with a positive correlation between maternal sera and cord blood concentration (Rho = 0.7669, *p* < 0.0001, Figure 2A). Bland–Altman plots of the difference between S1 receptor binding domain IgG antibodies in maternal sera and cord blood against maternal level are presented in Figure 2B (*p* < 0.001). Thirteen women received reduced-antigen-content diphtheria-tetanus-acellular pertussis vaccine (Tdap) administration in the third trimester. There was no difference in the percentage of inhibition of neutralizing antibodies between the Tdap group and the non-Tdap group (*p* = 0.5609, Figure 3).

## 4. Discussion

In this study, we measured SARS-CoV-2 neutralizing antibodies for wildtype and Delta variant in 29 maternal–neonatal pairs following maternal Moderna mRNA-1273 vaccination. Women who received one dose of vaccine had lower neutralizing antibody inhibition percentage compared with those that received two does (40.32% versus 97.46% in maternal sera, 43.33% versus 97.37% in cord blood, Table 2). According to the results of our study, the cord-to-maternal ratio of neutralizing antibodies is approximately one in fully vaccinated women (Table 2). Therefore, it can be expected that the fetus would obtain protection from passive immunity of transplacental maternal antibodies. IgG antibody level was also consistent with the results of neutralizing antibodies. The S1 receptor binding domain IgG antibody level in the one-dose group was much lower than that in the two-dose group. There were no differences between maternal and cord-blood S1 receptor binding domain IgG antibody levels. The current results indicate that full maternal immunization has the potential to maximize transplacental antibody transfer to offer adequate seroprotection in young infants. For the Delta variant, the effectiveness of existing vaccines is significantly reduced (Figure 1 & Table 2) [28]. Our results showed a near absence of maternal and cord-blood neutralizing antibodies for the Delta variant in the one-dose group (Table 2), which is consistent with previous findings indicating that a single dose of Pfizer-BioNTech or AstraZeneca either showed low or no efficiency against the Beta and Delta variants [29]. After a two-dose vaccination, our data indicates that the neutralizing antibodies against the Delta variant significantly increased at the end of two weeks but that level was lower when compared to wildtype (Figure 1). The results of the study are consistent with epidemiological data in the real world [30]. Our results are in accordance with previous reports evaluating the effectiveness of one-dose and two-dose immunizations, showing full vaccination along with clinical outcome. In our study there was no significant difference between the relationship of Tdap vaccination and inhibition percentage of neutralizing antibodies for COVID-19 (Figure 3), supporting the recommendation for COVID-19 vaccine administration during pregnancy.

A previous retrospective cohort study analyzed 503,875 individuals and demonstrated 51% vaccine effectiveness of Pfizer-BioNTech BNT162b2 vaccine in days 13–24 after one dose [31]. Another nationwide study indicated that estimated vaccine effectiveness during days 14–20 and days 21–27 after one dose was 46% and 60% for infection, and 57% and 66% for symptomatic COVID-19 [31]. Ida et al. observed vaccine effectiveness for healthcare workers in long-term care facilities was even lower (16%, 14 days after one dose and until second dose) [32]. All studies observed high-level protection against SARS-CoV-2 infection after a second mRNA vaccine immunization [32,33,34,35]. These results support the recommendation for a two-dose schedule of SARS-CoV-2 mRNA vaccines for adults without contraindications and previous severe adverse events as a means of providing maximum benefit. For pregnant women, the vaccine’s value is not only to protect themselves. Acquired passive immunity provides natural protective effect for young infants against disease. Our study highlighted that the concentration of neutralizing antibodies in the umbilical cord blood reached its highest value 4–8 weeks after maternal complete vaccination, and this result is particularly significant for the Delta variant.

The current global threat to vaccine strategy, immunity propagation, and controlling the spread of disease is virus variants. Although there are variations in different regions, the B.1.617.2 strain was predominant and outcompeted other variants in most countries. The diminishing level of neutralizing antibodies may explain why the Delta variant is more transmissible and why it is less affected by immune protection. The correlation between time from vaccination and breakthrough infection has been studied. A longer time period is associated with decreased vaccine-induced immunity [36] and increased risk of breakthrough infection [37].

For the pregnant women we studied, infection with the Delta variant resulted in a higher proportion of severe–critical disease and preterm births [38,39]. Regarding breakthrough infections from the Delta variant, fully vaccinated pregnant recipients had mild symptoms compared to unvaccinated women [37]. No specific type of vaccine can protect all recipients, but those in use have proven effective for reducing the risk of COVID-19 infection, symptom severity, and hospitalization for all variants [40,41,42,43]. Our research results showed that even though pregnant women had lower levels of antibodies to the Delta variant after being vaccinated, they should still have partial protection from disease infection. Our results highlight the maternal immunity effect of COVID-19 vaccination and the promising benefit of passive immune transfer to the fetus that reinforces the recommendation for vaccination during pregnancy, especially when Delta variant infections are surging. According to recommendations of American College of Obstetricians and Gynecologists (ACOG), all pregnant women should receive a Tdap vaccine during each pregnancy and an annual influenza vaccine during influenza season [44].

However, there is a lack of information to prove that recommended vaccines administered during pregnancy will not affect the effectiveness of the COVID-19 vaccine, obstetrician-gynecologists and other obstetric caregivers have declined to recommend routine vaccination to pregnant women who have been received COVID-19 vaccine. Our research shows that administering both COVID-19 and pertussis vaccines during pregnancy did not reduce the immunity response of COVID-19 vaccine. This result could offer obstetrician-gynecologists and obstetric care providers more convincing information when making vaccination recommendation for pregnant women.

The strengths of our study are the interested group of participants and neutralizing antibody to evaluate their immune responses. There are still very limited reports on the immune response of pregnant women after COVID-19 immunization. We evaluated and compared the level of neutralizing antibodies against wild type and Delta variants of SARS-CoV-2 produced by pregnant women after vaccination. Our study also disclosed the efficiency of the transferring antibodies produced by the mother to the fetus through the placenta. Since 18 December 2020, the Food and Drug Administration issued an Emergent Use Authorization (EUA) for the Moderna COVID-19 vaccine, the Center for Disease Control and Prevention recommended that pregnant and lactating women have access to the available COVID-19 vaccines. However, pregnant women were excluded from phase-three vaccine efficacy trial, thus data on vaccine efficacy in these populations remains limited. Concerned with the potential for severe illness from SARS-CoV-2 infection during pregnancy, the ACOG recommended that pregnant and recently pregnant women up to 6 weeks postpartum, including pregnant and recently pregnant health care workers, receive a booster dose of COVID-19 vaccine following the completion of their initial COVID-19 vaccine or vaccine series [45]. Regarding the vaccination timing in pregnancy that the booster dose should be administered to achieve the maximum benefit, like pertussis vaccination protocol, further research is needed. Our study indicated the levels of neutralizing antibodies in maternal sera and cord blood were high for 2–10 weeks after vaccination and this result might provide a reference for future booster dose vaccination.

On 24 November 2021, South Africa reported the identification of a new SARS-CoV-2 variant, B.1.1.529, which was later named Omicron, to the World Health Organization (WHO) [46]. In just one month, the Omicron variant caused another surge of infections worldwide. Omicron variant has become the main coronavirus strain in many countries, including the United Kingdom and the United States, at the end of 2022. Consecutive evidence shows that Omicron mutant has higher transmissibility than Delta mutants [47,48], and data from South Africa show that Omicron variant increased the risk of reinfection [49]. Vaccination is expected to protect against severe illness, hospitalizations, and deaths due to infection with the Omicron variant [47]. However, breakthrough infections in people who are fully vaccinated are likely to occur. Therefore, booster vaccination has been widely discussed recently. According to the Pfizer and BioNTech companies’ preliminary data reported in December 2021, a third dose induced higher neutralizing antibodies to Omicron, which may help to provide better protection [50]. The ACOG also recommended that all eligible pregnant and recently pregnant individuals receive a COVID-19 booster shot [51]. We will continue to follow the immune response of pregnant women after booster dose administration.

This study has several limitations. First, the study size is small, and most participants were vaccinated in the third trimester, and thus conclusion about appropriate timing for better vaccine efficacy has not been determined. Second, the comparison of immunogenicity post COVID-19 vaccination between pregnant and nonpregnant women was not evaluated. Third, for participants who received both COVID-19 and pertussis vaccines, we measured the neutralizing antibodies of SARS-CoV-2, not Tdap-related antigen specific antibodies, so it is impossible to know whether the effectiveness of the pertussis vaccination was affected.

## 5. Conclusions

Immunizing pregnant women is crucial to protect mothers and their offspring from infection. Given the high risk that COVID-19 imposes in pregnancy and reduced neutralizing antibody levels after one dose of mRNA vaccine, pregnant women should receive full vaccination to elicit a high immune response. Lower herd immunity to the Delta variant threatens vaccine strategies; however, the two-dose vaccination still provides a certain degree of protection and decreases severe disease in the real world. Before new vaccines are available, following the existing vaccination guidelines is the best policy. Further studies that examine the interaction between COVID-19 vaccine and other vaccine immunization during pregnancy are necessary.

## Figures and Tables

**Figure 1 vaccines-10-00101-f001:**
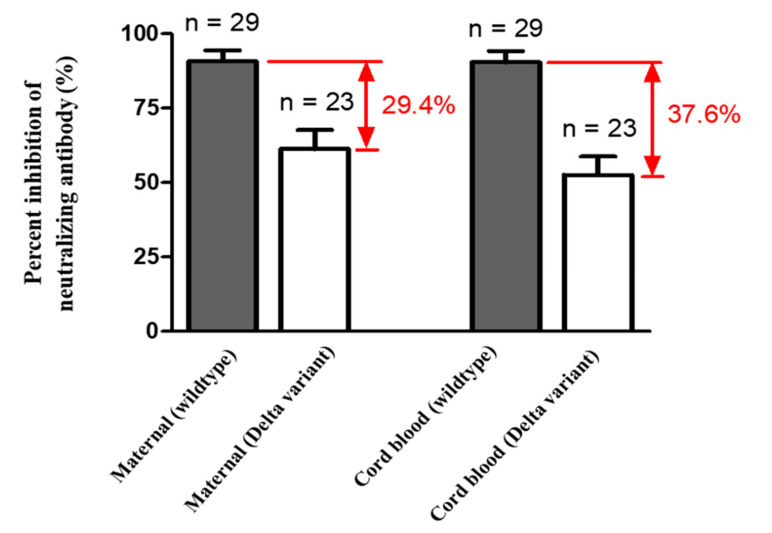
Mean percent inhibition of neutralizing antibodies in maternal and cord blood (wildtype *n* = 29 with two dose *n* = 25, one dose *n* = 4; Delta variant *n* = 23 with two dose *n* = 19, one dose *n* = 4; *p* value of maternal (wildtype) versus maternal (Delta variant) = 0.0001; *p* value of cord blood (wildtype) versus cord blood (Delta variant) <0.0001).

**Figure 2 vaccines-10-00101-f002:**
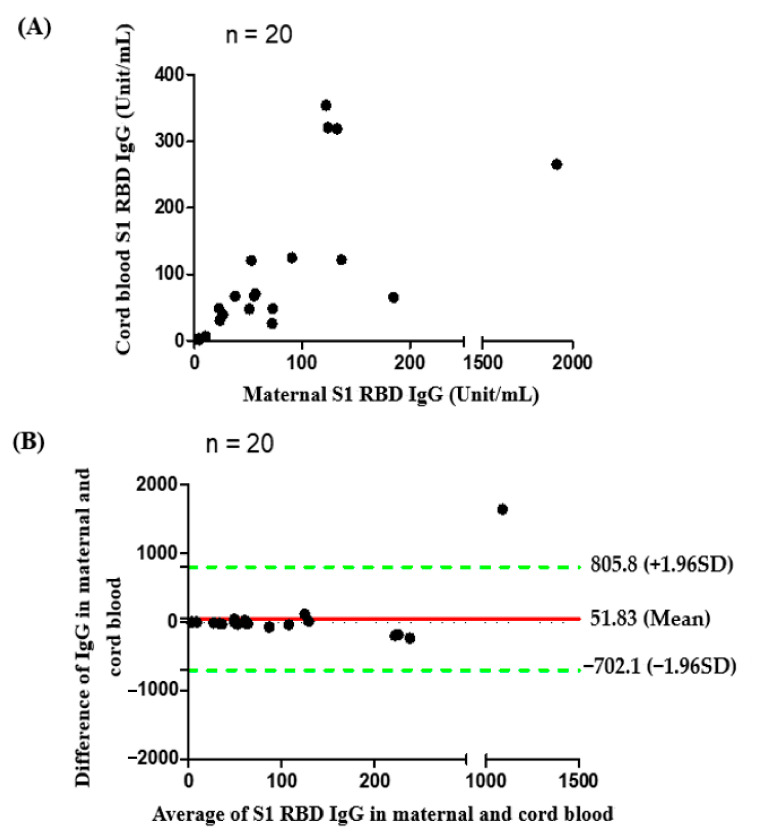
(**A**) Correlation of S1 receptor binding domain IgG antibodies between maternal and cord blood (*n* = 20, with two dose *n* = 16, one dose *n* = 4 Rho = 0.7669, *p* value < 0.0001); (**B**) Bland and Altman plot. The difference of S1 receptor binding domain IgG antibodies in maternal and cord blood in relation to the average of the two (*n* = 20, with two dose *n* = 16, one dose *n* = 4). Green lines indicate the limits of agreement (±1.96 standard deviation).

**Figure 3 vaccines-10-00101-f003:**
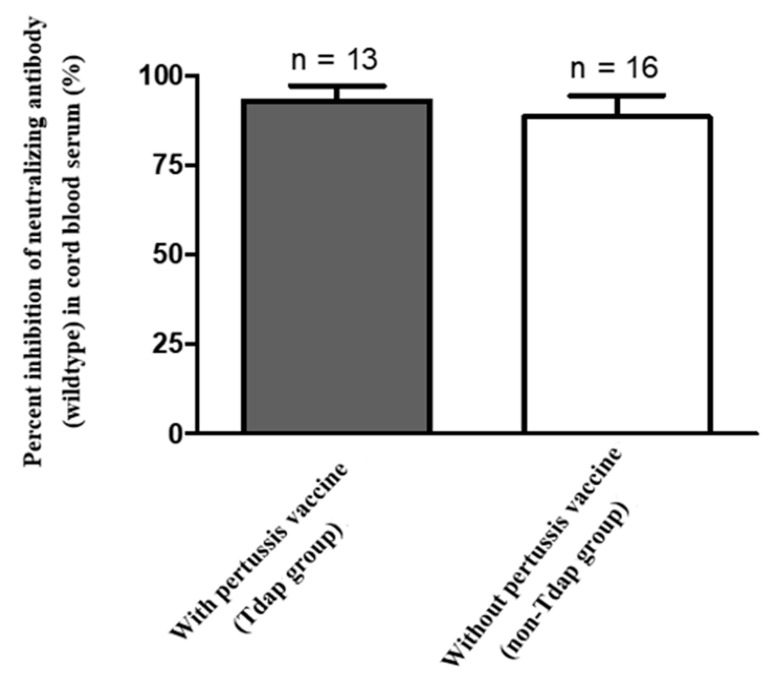
Mean percent inhibition of neutralizing antibodies (wildtype) in cord blood (*n* = 29, *p* value = 0.5609).

**Table 1 vaccines-10-00101-t001:** Maternal and newborn demographic and clinical data.

Variable	Included in the Analysis
Age of mothers (years) ^a^	33.21 * (±3.89 **) IQR 35–31
Parity ^a^	20 (68.97% ***) IQR 1–0
≥1	
BMI ^a^	27.42 * (±4.47 **) IQR 29.6–24.3
Weeks of gestation at the first dose of COVID-19 vaccination (weeks) ^a^	28.45 * (±2.64 **) IQR 30–27
Weeks of gestation at the second dose of COVID-19 vaccination (weeks) ^b^	33.31 * (±2.13 **) IQR 34–32
Interval between the second dose of COVID-19 vaccination and the collection of blood samples (day of delivery) (weeks) ^b^	5.23 * (±2.04 **) IQR 7–4
Interval between the first dose of COVID-19 vaccination and the collection of blood samples (day of delivery) (weeks) ^a^	10.00 * (±2.45 **) IQR 11–9
Weeks of gestation at delivery (weeks) ^a^	38.55 * (±1.12 **) IQR 39–38
Sex of newborn ^a^ Male Female	11 (37.93% ***) 18 (62.07% ***)
Weight of newborn (g) ^a^	3105.69 * (±357.07 **) IQR 3360–2900

BMI: body mass index; * mean; ** standard deviation (±SD); *** percentage of all surveyed patients: ^a^ case number = 29; ^b^ case number = 25.

**Table 2 vaccines-10-00101-t002:** Summary of neutralizing antibodies (wildtype and Delta variant) in maternal and cord blood.

Characteristics	Maternal Neutralizing Antibody (Wildtype)	Cord Blood Neutralizing Antibody (Wildtype)	Cord to Maternal Ratio (Wildtype)	Maternal Neutralizing Antibody (Delta Variant)	Cord Blood Neutralizing Antibody (Delta Variant)	Cord to Maternal Ratio (Delta Variant)	Rho ^a^	Rho ^b^	Rho ^c^	Rho ^d^
One-dose group (median, IQR)	40.32% * (51.74–25.28)	43.33% * (43.74–28.80)	1.07 * (1.23–0.89)	4.01% * (7.51–0.87)	1.44% * (2.16-)	0.92 * (1.40–0.86)	_	_	_	_
Two-dose group (median, IQR) (Interval of Second administration to delivery)	97.46% ** (97.73–97.18)	97.37% ** (97.59–97.05)	0.99 ** (1.00–0.99)	80.49% *** (85.50–55.78)	66.25% *** (75.22–44.68)	0.90 *** (0.95–0.78)	0.5681 **	0.2389 **	0.2963 ***	0.2343 ***
0–2 weeks	95.99% ^1^ (96.41–95.58)	95.78% ^1^ (96.38–95.18)	0.99 ^1^ (1.01–0.99)	49.96% ^1^ (51.06–48.87)	41.86% ^1^ (53.26–30.46)	0.86 ^1^ (1.11–0.61)	_	_	_	_
2–4 weeks	97.45% ^2^ (97.61–97.23)	97.16% ^2^ (97.40–96.57)	0.99 ^2^ (1.00–0.99)	80.72% ^3^ (84.74–62.42)	57.48% ^3^ (63.47–47.82)	0.78 ^3^ (0.90–0.75)	_	_	_	_
4–8 weeks	97.48% ^4^ (97.74–97.30)	97.57% ^4^ (97.76–97.19)	1.00 ^4^ (1.00–1.00)	81.76% ^5^ (87.51–68.61)	70.60% ^5^ (80.78–61.16)	0.92 ^5^ (0.95–0.87)	_	_	_	_
8–10 weeks	97.72% ^6^ (97.72–97.72)	97.30% ^6^ (97.30–97.30)	0.99 ^6^ (0.99–0.99)	_	_	_	_	_	_	_

^a^—rho of cord-blood neutralizing antibody (wildtype) and number of weeks from vaccination to delivery; ^b^—rho of cord-to-maternal ratio (wildtype) and number of weeks from vaccination to delivery; ^c^—rho of cord-blood neutralizing antibody (Delta variant) and number of weeks from vaccination to delivery; ^d^—rho of cord-to-maternal ratio (Delta variant) and number of weeks from vaccination to delivery; * case number of one-dose group = 3; ** case number of two-dose group = 26; *** case number of Delta variant group = 20; ^1^, case number = 2; ^2^, case number = 7; ^3^, case number = 6; ^4^, case number = 16; ^5^, case number = 12; ^6^, case number = 1.

## Data Availability

The datasets of this research are available on request to the corresponding author.

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
