# Peer review of "Evaluation of Transplacental Antibody Transfer in SARS-CoV-2-Immunized Pregnant Women"

_vaccines, 2022, doi:10.3390/vaccines10010101_

Round 1

Reviewer 1 Report

Comments on the manuscript: “Evaluation of transplacental antibody transfer in SARS-CoV-2-immunized pregnant women” by Ching-Ju Shen et al.

Authors present results from a prospective studywhere 29 pregnat women were enroled and received 1 or 2 doses of Moderna mRNA vaccine, antibody titres and serum neeutralising activity vs wild type and Delta variant, were evaluated from blood samples from 29 mothers and from the umbilical cord blood of 29 babies on the day of delivery.Thirteen women received antigen reduced diphtheria-tetanus-acellular pertussis vaccine in the third trimester that had no impact on the neutralizing antibody response compared to group that not received this vaccine. Two doses induced anti-RBD antibody titres and neutralisation activity in the mother similar to the reported in the literature and were superior to the response induced by one dose. Anti-RBD antibody titres and neutralising activity was also observed in the cord blood of the babies, however not at the same level as in the mothers, one dose of the vaccine did not induced neutralising activity in the cord blood of the babies. Antibody titres and neutralising activity against delta variant was reduced in all cases. The vaccination with antigen reduced diphtheria-tetanus acellular vaccine did not impact the anti-RBD and sera neutralising activity in the vaccinated woman and in the babies cord blood.

The manuscript is clearly presented; results are consistent with the author`s interpretations and contribute with relevant data to understand antibody responses induced by mRNA COVID-19 vaccines in pregnant women and their babies, and that using other vaccines during pregnancy may not affect the responses induced by the COVID-19 vaccine.

  1. Please include anti-N antibodies titres in the mothers and in the cord blood to show that the response observed was not due to a previous asymptomatic infection. Since PCR was only taken at enrolment there is not proof that the mothers were not infected previously (even before pregnancy)
  2. Please include a table of secondary effects studied in mothers and babies. Safety and tolerability information will importantly benefit and strength manuscript conclusions.

3. Please include the number of women that received only one dose of the vaccine, include the information of neutralisation by one dose in the figure 1 and a figure comparing anti-RBD IgG responses in the study groups.

Reviewer 2 Report

In this study, the authors reported the percentage level of maternal and umbilical cord blood neutralizing antibodies against Sars-Cov-2 Spike protein developed after vaccination. The results demonstrated the importance of the full vaccination in pregnancy and lower levels of neutralizing antibodies for delta variant.
it is an interesting study that adds an important piece to the puzzle of the COVID-19 challenge. 
I suggest adding a comment about the comparison between the neutralizing levels developed by pregnant women and non-pregnant ones and to empathize that the authors failed to describe the occurrence of adverse effects.
Furthermore, in my opinion, the M&M section is too detailed: the entire protocol used for a commercial ELISA kit is not needed.
I suggest to add a review article about the complication and the pathogenetic mechanisms of COVID-19 in pregnancy (https://doi.org/10.3389/fimmu.2021.775168)

Reviewer 3 Report

Dear Authors,

Although the number of individuals in the study is relatively small, the results obtained are important. I have a few comments that should be noted in the manuscript.

Line 149. The number of women with one or two doses should be part of 2.1. (Study Design and Patients) instead of Results.

-3.2. Sample Characteristics, lines 169-177. The authors described the results in women vaccinated with one or two doses. Please specify in the text the strain against which neutralising antibodies are measured. 

- Line 183. I disagree with the sentence "The median cord-to-maternal ratio of Delta variant neutralizing antibody in the two-dose group did not differ in regard to vaccination timing.". The ratio varies, decreasing for 2-4 weeks, and increasing again between 4-8 weeks. Does those differences statistical significant?. 
Moreover, for 2-4 weeks the antibodies in cord-blood increased 68%. This could be highlighted in results and discussion

- Line 183. I disagree with the sentence "The median proportion of neutralising antibodies to the Delta variant in the two-dose group did not differ from the time of vaccination". The proportion varies, decreasing over 2-4 weeks, and increasing again between 4-8 weeks. Are these differences statistically significant? 
Furthermore, during 4-8 weeks antibodies in cord blood increased by 68%. This could be illuminated in the results and discussion.

- Line 220. The number of women are 28 or 29?

- I suggest that the authors should give more importance in the Discussion that 2 doses after 4-8 weeks after 1st dose greatly increases the titer of neutralising antibodies in cord blood.

Round 2

Reviewer 1 Report

Authors duly response the comments raised to the manuscript and modified accordingly.

Author Response

Thank you for this great comment from the reviewer!